# Data-Driven Technology Roadmaps to Identify Potential Technology Opportunities for Hyperuricemia Drugs

**DOI:** 10.3390/ph15111357

**Published:** 2022-11-03

**Authors:** Lijie Feng, Weiyu Zhao, Jinfeng Wang, Kuo-Yi Lin, Yanan Guo, Luyao Zhang

**Affiliations:** 1Logistics Engineering College, Shanghai Maritime University, 1550 Haigang Avenue, Pudong District, Shanghai 201306, China; 2Institute of Logistics Science and Engineering, Shanghai Maritime University, 1550 Haigang Avenue, Pudong District, Shanghai 201306, China; 3China Institute of FTZ Supply Chain, Shanghai Maritime University, 1550 Haigang Avenue, Pudong District, Shanghai 201306, China; 4School of Business, Guilin University of Electronic Technology, Guilin 541004, China; 5School of Life Sciences, Shanghai University, Shanghai 200444, China; 6School of Life Sciences, Zhengzhou University, No. 100 Science Avenue, Zhengzhou 450001, China; 7School of Economics and Management, Shanghai Maritime University, 1550 Haigang Avenue, Pudong District, Shanghai 201306, China; 8School of Computer and Information Engineering, Henan University of Economics and Law, Zhengzhou 450016, China

**Keywords:** data-driven TRM, SAO analysis, link prediction, hyperuricemia drug, human uric acid oxidase

## Abstract

Hyperuricemia is a metabolic disease with an increasing incidence in recent years. It is critical to identify potential technology opportunities for hyperuricemia drugs to assist drug innovation. A technology roadmap (TRM) can efficiently integrate data analysis tools to track recent technology trends and identify potential technology opportunities. Therefore, this paper proposes a systematic data-driven TRM approach to identify potential technology opportunities for hyperuricemia drugs. This data-driven TRM includes the following three aspects: layer mapping, content mapping and opportunity finding. First we deal with layer mapping. The BERT model is used to map the collected literature, patents and commercial hyperuricemia drugs data into the technology layer and market layer in TRM. The SAO model is then used to analyze the semantics of technology and market layer for hyperuricemia drugs. We then deal with content mapping. The BTM model is used to identify the core SAO component topics of hyperuricemia in technology and market dimensions. Finally, we consider opportunity finding. The link prediction model is used to identify potential technological opportunities for hyperuricemia drugs. This data-driven TRM effectively identifies potential technology opportunities for hyperuricemia drugs and suggests pathways to realize these opportunities. The results indicate that resurrecting the pseudogene of human uric acid oxidase and reducing the toxicity of small molecule drugs will be potential opportunities for hyperuricemia drugs. Based on the identified potential opportunities, comparing the DNA sequences from different sources and discovering the critical amino acid site that affects enzyme activity will be helpful in realizing these opportunities. Therefore, this research provides an attractive option analysis technology opportunity for hyperuricemia drugs.

## 1. Introduction

The amount of uric acid in the body needs to be kept at a stable level. While the synthesis of uric acid increases or the amount of uric acid excreted from the body decreases, the concentration of uric acid in the blood increases [1,2,3]. A person is considered to have hyperuricemia when the uric acid level in the blood exceeds the average levels [4,5,6,7]. The increased intake of high-fat, high-protein, and high-sugar food leads to metabolic disorders and a rising risk of hyperuricemia [8,9,10]. It is estimated that there are currently about 17.7 million hyperuricemia patients worldwide. As can be seen, hyperuricemia positively correlates to many other potential diseases, such as obesity, hypertension, diabetes, cardiovascular disease, and chronic kidney disease [11]. The primary approach in treating hyperuricemia is to rebalance uric acid synthesis and excretion. Drugs for treating hyperuricemia are divided into three categories: xanthine oxidase inhibitors, urate anion transporter 1 (URAT1) and urate oxidase. However, most of these chemical drugs for hyperuricemia have specific side effects, and the oxidase-based drugs produce antibodies with long-term use. Neither of them can dissolve uric stones deposited in the joints [12,13]. Technological innovation for hyperuricemia drugs is imperative.

The interest in developing hyperuricemia drugs is continuously growing. Monitoring hyperuricemia pharmaceutical technologies are fundamental to analyzing and identifying potential technology opportunities. It is helpful to narrow the scope of technology research topics and reduce R&D risks, which not only supports the decision-making of hyperuricemia drug research but also helps to avoid unnecessary R&D costs [14,15,16,17]. Various data analysis tools were explored to analyze potential technology opportunities efficiently [18,19,20], such as bibliometrics [21], citation analysis [22], technology roadmap (TRM) [23], Biterm Topic Model (BTM) [24], Bidirectional Encoder Representation from Transformers (BERT) [25,26], Subject-action-object analysis (SAO), [27] and link prediction [15,28,29]. These tools integrate mathematics, statistics, computer science, and operations research in technology opportunity analysis [16,30,31,32,33]. However, these tools do not work independently, but often combine into a new, more efficient analysis path.

TRM is a comprehensive approach to capturing changes in technology and markets over time in an integrated manner. It is not only a flexible approach to analyzing technologies and market requirements [34,35], but it can also create a more effective way to track and analyze the latest technology trends by integrating data analysis tools [36,37,38]. Pharmaceutical technology innovations are influenced by various factors, such as changing customer expectations, uncertain intellectual property (IP) procedures, unconsidered technology changes, and resource requirements [23,39,40,41]. It is necessary to use TRM to analyze technology opportunity trends from both technological and market dimensions. In addition, TRM is a valuable tool in shortening the technology development cycle, discovering drug targets, and optimizing resource consumption. There are three major categories of research on TRMs: theory-based, case study-focused, and data/methodology-specific [15]. Among numerous extensions of TRM, data integration is a notable trend. Some researchers have employed the data-driven approach in TRM. For example, Yu developed a patent roadmap for the competitive market and patent layout planning analysis [42]. Zhou traced the innovation path of Solid lipid nanoparticles [37].

Despite the contributions of previous research using TRM to analyze TOA, data-driven TRM have some limitations in three areas: processing data data-driven TRM, identifying potential opportunities, and selecting the data source. From the data processing standpoint, most previous research has adopted keyword-based network analysis approaches for data-driven TRM. However, traditional keyword-based analysis approaches neglected to express the relationships between technologies and the market, inadequately reflecting the contexts. Compared with identifying potential technology opportunities, most researchers relied on current trends from the perspective of technology forecasting. The technology opportunity analysis (TOA) process contains six main stages. It includes data acquisition, description, potential relationship extraction, visualization, analysis of results, and identifying potential technical opportunities. Instead of identifying potential technology opportunities, the current TRM focuses on the first five steps in TOA. Technology opportunities are primarily based on technology hotspots, while technology opportunities often exist in potential connections. From the perspective of data source selection, technology and the market have become complex. Some critical technical information exists not only in patents, but also in literature and market reports. Similarly, market information is hidden not only in market reports but is also embedded in patents and literature. Therefore, it is necessary to construct a comprehensive data-driven TRM to automatically, quickly, and accurately extract technology and market data from a large amount of literature, patent, and commercial data.

Therefore, this article is proposing a systematic method for developing data-driven TRM to identify potential technology opportunities for hyperuricemia drugs which contains three stages. The first stage is layer mapping. We map the literature, patent, and commercial data into the technology and market layers based on BERT and semantic analysis for the technology layer and market layer based on SAO. The second is content mapping. We identify topics of SAO components for technology and market layers based on BTM. The last stage is opportunity finding. We identify possible links between unconnected nodes based on link prediction. This data-driven TRM effectively identifies potential technology opportunities for hyperuricemia drugs and suggests pathways to realize these opportunities.

The rest of this paper is organized as follows. Section 2 describes the relevant thermotical background on TOA, TRM, and data-driven TRM regarding BTM, SAO, and link prediction. Section 3 outlines our proposed approach for a data-driven TRM. It explains integrating BTM, BERT, SAO, and link prediction to analyze technological opportunities for hypouricemic drugs. A case study of technology prediction related to hyperuricemia is then presented in Section 4. Section 5 discusses our discovery and extension of the study. Section 6 summarizes the paper, looks at possible future research, and provides some limitations of our study.

## 2. Theoretical Background

### 2.1. Technology Opportunity Analysis

Technology opportunity analysis (TOA) helps researchers and organizations explore potential technological opportunities. It also enables a better understanding of scientific and technological developments by deeply mining valid information in publications, patents, and the literature [43].

Many researchers have developed effective methods to identify and predict technology opportunities. Early research used qualitative analysis methods that relied on expert experiences, such as Delphi and Workshop. While in specialized fields, specialist opinion can provide creative foresight for analyzing technology opportunities. However, information increased steeply. It is impossible to consistently identify technology trends based on expert knowledge alone, which is time-consuming and costly. In addition, specialist judgment is often limited by personal expertise and bias. Sometimes consensus cannot be reached.

Bibliometrics was first introduced to analyze technological opportunities for emerging technologies [44], which is used to evaluate R&D activities by counting the number of authors and literature citation relationships [45,46]. This method has been widely used for the analysis of technology evolution trajectories in energy [47], conductive polymer nanocomposites [48], and robotics [49]. Bibliometrics provides quantitative data and objective evidence to evaluate technical opportunities and reach a consensus of experts. However, bibliometrics cannot extract the meaning of documents in-depth but can only reflect information such as the flow of knowledge, citations of literature, and patents. However, it has a time lag.

Then text mining is then introduced in TOA, which is suitable for unstructured text data analysis and can extract text features in-depth [50]. There are many data analysis techniques such as machine learning and natural language processes arose. Some research has focused on developing automated and semi-automated data analysis methods in TOA [51,52,53]. Among them, principal component analysis (PCA) and text clustering are often used to extract topic information [37]. The similarity is used to measure connections between technical topics [54]. For example, Wu predicted evolutionary relationships between stem cell themes based on LAD, HMM, and co-occurrence theory [55]. Du used the topic models to predict potential topics for new drugs [22]. Zheng presented text mining tools to reveal possible innovation pathways and commercial applications of solid lipid nanoparticle drugs [56]. Zheng reviewed the importance of machine learning in facilitating the translation of bioenergy and biofuel innovations [57].

TOA has evolved over a long period of time and has been enriched by many scholars. To analyze potential technology opportunities efficiently, various data analysis tools were explored. These tools integrate mathematics, statistics, computer science, and operations research in TOA. However, rather than working independently, these methods are often combined into a new, more efficient analysis path.

### 2.2. Technology Roadmap and Data-Driven Technology Roadmap

TRM is a time-based multi-layer chart that can be integrated with various data analysis tools to form a more efficient analysis path in TOA. TRM-based technology opportunity analysis can better identify the dynamic distribution of technology. It can also predict technology development trends and identify potential technology opportunities in a time series [34,40,58]. It usually consists of three layers: the market, product, and technology [39,59] layers, as shown in Figure 1. The existing research of TRM in TOA is mainly categorized into the following streams: theory-based, case study-focused, and data/methodology-specific.

Regarding theory-based TRM, some previous studies have focused on the concept and process of TRM [60,61]. To support market-pull and technology-driven innovation, new frameworks for TRM have been proposed, such as T-Plan TRM [62], learning-based TRM [63], or umbrella-based TRM [61].

In terms of a case study-focused on TRM, some previous studies have focused on applying TRM in different industries or sectors [59,64,65,66,67]. To accommodate the domain-specific and case-specific needs in other areas, the customization of TRMs has been developed. There are TRMs in the aeronautical and aerospace sectors [68] and in robotics technologies in the power sector [69]. There are also TRMs in agile hardware development [67] and pharmaceutical technology landscape development [70].

Regarding data/methodology specific to TRM, some previous studies have focused on integrating data analysis tools to develop an efficient TRM [58,71,72,73]. The TRM has excellent flexibility in the structure and development process. Various tools can be flexibly selected to build a TRM according to different purposes of TOA. For example, researchers have integrated various tools into TRM to accommodate the complex business environment and the rise of big data. Means such as technology mining (TM), analytic hierarchy process (AHP) [74], business model canvas (BMC) [75], cross impact analysis (CIA) [76], and fuzzy set theory [58] have been employed. Some studies used technology mining-based patents analysis for technology roadmaps to explore AI-healthcare innovation [77]. Furthermore, some studies employ tools such as Bayesian network and topic modeling to develop a risk-adaptive technology roadmap under deep uncertainty [78].

With the rise of big data analytics and the rapid change in the business environment, TRM integrating data and analysis methodology for TOA is becoming increasingly popular. More and more researchers concentrate on the importance of data to TRM [79]. The data-driven technology roadmap is gradually proposed [80]. The data source for data-driven TRM is increasingly diversified, mainly including patents, literature, and commercial data. Data analysis tasks can be used for data-driven TRM, such as text classification, summarizing, key information extraction, topic clustering, semantic analysis, navigation, topic visualization, and node linking [15,71,81]. To put the data source selected into proper layers of TRM, some studies choose data analysis tasks, such as text classification models [82]. Some studies employ data analysis tasks to identify potential technology topics for TRM, such as text clustering tools [57,83]. Some studies used semantic analysis tools to extract critical technology information, such as SAO [84,85]. And to identify potential technology opportunities, some studies adopt link prediction [15].

### 2.3. Data Analysis Techniques and Data-Driven Technology Roadmap

Data analysis techniques have been increasingly employed to support quantitative and intelligent data-driven TRM. 

#### 2.3.1. Bidirectional Encoder Representations for Transformers with the Data-Driven Technology Roadmap

Data analysis tools, such as text classification models, can be used to put the data source selected into proper layers of data-driven TRM. Classification models such as support vector machine (SVM) [86], k-nearest neighbor (KNN) [87,88], Hidden Markov [89], and Bayesian [44,90] can be employed. 

To improve classification accuracy, these text classification models should be trained based on a massive manually labeled training dataset [86], which is time-consuming, labor-intensive, and costly. The low accuracy of the classification model is often caused by the small sample size, inefficient model computation, and high reliance on domain experts. To improve the accuracy of classification based on small sample training set models, the BERT model is proposing in 2018. The model has been widely used for its excellent performance in text classification [91]. When dealing with domain-specific classification tasks such as pharmaceutical technology, it is required to construct small domain sample datasets and pre-train the model [17,92,93]. 

This paper intends to create a small domain-specific training set. This will be followed by training the BERT model based on fine-tuning [94] to accurately classify pharmaceutical data into proper layers of data-driven TRM.

#### 2.3.2. Subject-Action-Object Analysis with the Data-Driven Technology Roadmap

Data analysis tools, such as SAO, can extract critical technology information for layer mapping of data-driven TRM [95]. Initially, the SAO structure was widely used to analyze technical documents such as patents, present valid technical information, critical technocratic findings, and represent the relationships between technical elements [96,97,98,99,100,101,102]. Guo constructed SAO chains to identify future directions of technology [103]. Wang identified technology opportunities based on SAO and the morphological matrix [104,105]. Natural language processing techniques have enabled SAO structures to express rich semantic information compared with topics. Therefore, it is considered an effective tool for identifying critical technical inter-elements in a corpus [106].

Subsequently, the SAO structure has been extended to many other fields, such as patent similarity analysis [85,107] and patent network analysis [108]. It can also apply to technology tree analysis [96,109], technology trend analysis [110], online review demand extraction [27], and M&A target selection [101].

Although SAO is widely used in TRM with TOA, there are still some limitations. When constructing the data-driven TRM, if the SAO structures are adopted directly without refining, the TRM is likely has a large amount of redundancy. It is inappropriate for efficient analysis and needs further refinement [111]. Therefore, it is necessary to identify topics of SAO components for different layers of data-driven TRM based on topic modeling tools [27,105].

#### 2.3.3. Biterm Topic Model with the Data-Driven Technology Roadmap

Data analysis tools, such as topic models, can identify potential technology topics for data-driven TRM [112,113]. Identifying topics of SAO components for different layers via a topic modeling tool can help researchers understand the target domain effectively. It can help to extract which areas the technical solution focuses on, how the solutions work, and which parts of the solution are the targets [44,52]. The most popular topic modeling techniques are LDA [81]. The LDA model has been widely used in various fields, such as text mining, bioinformatics, and image processing. The model has proven effective in extracting topics from large amounts of text data and analyzing technical topic changes [114,115].

However, the data is increasingly various, and short text data has emerged and been exploded. Sparse and unbalanced texts characterize short texts. The accuracy of extracting topics using the same topic modeling algorithms as long texts, such as LDA, is low. There is an urgent need to propose a topic model suitable for short texts [24]. Yan proposed a topic model algorithm, BTM, that is more suitable for short text clustering [116]. The model enhances the learning efficiency of the topic model by calculating the unordered co-occurrence word pairs. It effectively solves the semantic sparsity problem of short texts [117]. BTM can automatically extract hot and potentially technical topics from large amounts of short text data, even in the face of domain-specific datasets. BTM has become one of the most widely used short text modeling technologies [118].

BTM analyzes the technologies and market topics based on keywords that can’t reflect the contexts. To reflect the contextual semantics of the topics, BTM is more effective when combined with SAO [108]. However, the SAO structure consists of phrases. BTM is suitable for identifying topics of SAO components, which helps reduce the redundancy of SAO effectively. Some limitations exist in using BTM and SAO for technical opportunity analysis. They only consider the existing relationships and links, and pay less attention to identifying potential technology opportunities in automation [102]. However, technology opportunities possibly exist in potential connections [111]. SAO and BTM need to be combined with predictive tools for better performance in TOA, such as link prediction. 

#### 2.3.4. Link Prediction with Data-Driven Technology Roadmap

Data analysis tools, such as link prediction, can identify potential connections for data-driven TRM. Link prediction is a technique for discovering nodes or links in a network that are currently unknown but may be connected in the future. It has been well developed and applied in social network analysis and TOA [119,120]. There are three significant categories of link prediction: link prediction based on similarity, maximum likelihood estimation, and probabilistic models. Link prediction based on maximum likelihood estimation is unsuitable for massive amounts of data with low prediction accuracy. Link prediction based on probabilistic models often relies on external attributes of nodes, which are often difficult to obtain. In contrast, similarity-based link prediction is more accurate and is widely used [121].

With the development of classification models, Hansen found that link prediction can be constructed based on them [122]. Supervised classification models based on Bayesian, neural networks, and support vector machines (SVM) [123] can be employed to improve model accuracy. Subsequently, scholars began to compare the performance of link prediction based on classification models in processing domain datasets. In TOA, Yoon compared similarity-based and SVM-based link prediction performance. We can try to identify potential technology opportunities based on more classification models, such as Lightgbm with link prediction [123]. 

Link prediction is increasingly becoming a research hotspot in technology prediction. It has been widely used in the technical analysis of biological and medical patent data [124]. Shibata performed link prediction analysis in five large citation networks [125]. Xiao combined SAO with link prediction for identifying technical opportunities in skin melanoma [111]. Ma proposed a link-prediction-based technical knowledge network framework to predict potential technical opportunities in Alzheimer’s disease [126]. 

## 3. Methodology

This article proposed a systematic method for developing data-driven TRM to identify potential technology opportunities for hyperuricemia drugs. It contains three stages. The first is layer mapping. We classify the literature, patent, and commercial data into layers based on BERT and semantic analysis for the technology layer and market layer based on SAO. The second stage is content mapping. We identify topics of SAO components for technology and market layers based on BTM. The last stage is opportunity finding. We identify possible links between unconnected nodes based on link prediction. The data-driven TRM benefits technology needs assessment and technology response development in the technology roadmap process. The proposed model consists of three modules, as in Figure 2.

### 3.1. Collecting and Pre-Processing Data for Technology and Market

#### 3.1.1. Data Collection

The numerous databases of patents, academic papers, journals, and business reports contain voluminous technology and market information. Their abstracts are also stored in a structured database format, making them a beneficial source for data analysis. Since this study calls for technical and market-related data in developing a data-driven technology roadmap, we employ Medline, Derwent Innovations Index (DII), and Abstracts of Business Information (ABI) databases as data source collection. We then use the different search queries related to the research topic to download relevant scientific papers, patents and business journals and reports. 

#### 3.1.2. Setting the Timeframe of Data-Driven TRM

Considering the complex and dynamic technology replacement and market changes, one core of the TRM is setting the time frame. The technology opportunity analysis can be done within each time frame. Different rules, such as S-curve, can set the time frame. In the S-curve, the generation and development of technology have their pattern and trajectory. The stages of the technology life cycle are predictable and iterative. Building a time frame based on the S-curve helps to identify technology opportunities in a forward-looking manner. Therefore, this study uses the s-curve-based model to determine the development stages of R&D activities of technologies, and to identify technology and market trends within the altered time frame [127].

### 3.2. Layer Mapping

#### 3.2.1. Classifying the Data into Layers Based on BERT

In the second module, BERT, the text classification algorithm based on fine-tuning is used to classify the tech-related and market-related data into layers for data-driven TRM. The previous research on TRM using data-driven approaches tends to use abstracts. The abstract is an overview of the full text and facilitates a rapid discovery of high-value information with low-value density data.

This section consists of the following steps: First, the abstracts are separately extracted from the database in a timeframe. After that, we pre-process the tech-related and market-related data in text format in the timeframe. This article chooses the sent tokenize module of the Natural Language Toolkit (NLTK) in the Python package to divide the abstracts into sentences. We then conducted a BERT model to classify technical and market data. Even if BERT performs well in classification tasks based on the Google corpus, organizing technical and market data requires domain training sets, such as pharmaceutical data. To construct the domain training set, domain experts will invite, and 30 percent of sentences will be extracted randomly from the entire data set. The extracted training set will then be manually labeled with technology-related, market-related, and irrelevant data. After that, we will pre-train the BERT model. Only the technical-related and market-related data will be left. Finally, the whole dataset will be divided into several subsets and classified into technology and market layers of data-driven TRM in the timeframe.

#### 3.2.2. Semantic Analysis for the Technology Layer and Market Layer Based on SAO 

The SAO structures are a machine learning technique. It is always employed to obtain objective, structure, and effect data from text and then converts that information into structured text data. The SAO structures consider contextual meaning, which is superior to the keyword-based analysis. Thus, we chose the SAO technique to extract technical and market semantic structures, reflecting the contexts. 

This section consists of the following steps: first, we will extract the SAO structures. SAOs cannot be extracted without the help of a parser, which can analyze textual data through regular syntax rules. In this study, we employ the Stanford Parser, (Standford Parser, 3.9.2-models; package for extracting SAO structures) available as an open-source package for sentence separation [128]. Next, we extract the SAOs with a series of linguistic algorithms from each sentence in the timeframe. After that, we will filter, clean, and combine the SAO structures. The data used for technical analysis is likely to be very large with a low-value density. It contains all of the SAO structures. It is not appropriate for high-efficient analysis and needs to be filtered, cleaned, and combined. Therefore, we delete the duplicated technology and market SAO structures; only the unique SAOs are left. However, in the base of the dependency parser, the SAO (subject + action + object), SO (subject + action), and AO (adjective + object) structures all collected [129]. We remove the technology and market SAOs without subjects, actions, or objects, such as SO and AO.

### 3.3. Contents Mapping

#### 3.3.1. Pre-Processing the SAO Components

There will be vast redundancy if the contents mapping is solely based on SAO semantic analysis to identify potential technology opportunities. Hence, in this module, in addition to filtering the SAO structure above, we will employ a text clustering approach toward the dimensionality reduction of SAOs. Short text data is sparse text and an imbalance of data. Text clustering algorithms such as LDA cannot extract topics of short tests. Most of the SAO structures are phrases. We selected BTM, more appropriate for short text clustering, to extract technology and market SAOs components topics.

This section consists of the following steps: first, we will divide the remaining technology and market SAOs into several subsets. We will then remove the data noise and pre-process the sub-datasets group by group, such as word token, stemming, lemmatization, and excluding stop words. 

#### 3.3.2. Identify Topics of SAO Components for Technology and Market Layers Based on BTM

We conducted a BTM-based topic model to identify meaningful core and potential technology and market topics automatically. Perplexity is a crucial index to evaluate the clustering effect of topic modeling models. Nonetheless, perplexity cannot explain the semantic coherence of words for each topic on a non-probabilistic model. Topic coherence can describe it. Therefore, we chose coherence as a metric to evaluate the BTM model’s effectiveness.

This section consists of the following steps: first, we evaluate the value of coherence while varying the number of topics to determine the optimal number. Second, we pre-train the BTM model. Lastly, we identify topics of SAO components for technology and market layers based on BTM.

### 3.4. Opportunity Finding

#### 3.4.1. Identify Potential Connections Based on Link Prediction

The core of building a data-driven roadmap is to predict possible technology opportunities. However, content mapping only analyzes past data and ignores potential future technological opportunities. Therefore, in this module, we chose link prediction to predict the potential links between unlinked nodes. 

This section consists of the following steps: first, we select the results of SAO pre-processing in Section 3.3.1 as a train set. We then train the model based on link prediction to construct the overall network. Finally, we identify the probability of potential links for all unlinked topic nodes based on the trained link prediction model. 

#### 3.4.2. Integrating TRM and Analyzing Technology Opportunities

It is challenging to analyze such a large-scale network, so we only keep and interpret technology and market subnetworks. Subnetwork nodes are selected from the topics extracted by BTM. 

This section consists of the following steps: first, we map the technology and market subnetworks in a time series and layer series in two-dimensional data-driven TRM. Figure 3 shows an example of the final visualization. As shown in the figure, the technology roadmap is divided into two layers, technology and market. For example, the technical layer is divided into three sub-layers, from sub-layer S, sub-layer A to sub-layer O, along the vertical axis. The horizontal axis is the time series arranged in the timeframe—the same for the market layer.

Second, we select the nodes and edges in the technology and market subnetworks where potential links exist and visualize them in technical and market layers. If there is a potential link between two unlinked topics, the two nodes are linked with a line with arrows. The edges’ width and arrows represent the probability of a potential link between unconnected nodes. The wider the edges and arrows, the higher the likelihood of a potential link. We use these connections as possible directions for future technology and market development.

We then divide the link prediction visualization results into different communities in technical and market layers [130,131]. Arrow edges with different colors represent different community themes. Dashed cycles with different colors highlight diverse communities. 

Lastly, we analyze possible opportunities for future technology and market development based on the final technology roadmap.

## 4. Illustrative Example

### 4.1. Collecting and Pre-Processing Data for Technology and Market

#### 4.1.1. Data Collection

The dataset of this study was derived from three distinct databases extracted from Medline, Derwent, and ABI, respectively. We then used the Mesh term ‘MH = (gout OR hyperuricemia)’ as a search query from Medline. We chose the International Patent Classification Number ‘IP = (A61P-019/06)’ as a search query from Derwent. And we selected the keyword “hyperuricemia” as a search query for ABI. The cutoff date was 31 December 2021. Any data beyond that date are not part of this study. Therefore, a set of 6124 hyperuricemia-related essays, a collection of 5158 hyperuricemia-related patent data, and 4582 hyperuricemia-related commercial data were extracted, as shown in Table 1. The study keeps the dataset with abstracts. It includes 5066 literature abstracts, 5158 patent abstracts, and 1447 commercial data. The databases used in this study are presented in Table 1. The literature abstracts, patent abstracts, and commercial data were analyzed for technology opportunities.

#### 4.1.2. Setting the Timeframe of Data-Driven TRM

This study divides the part of the dataset with abstracts related to hyperuricemia drugs into three sub-periods according to the S-curve, as shown in Figure 4. They are the stable period (2010–2013), the rising period (2014–2018), and the fluctuating period (2019–2021), which are represented by TS_1_, TS_2,_ and TS_3,_ respectively. This study then analyzes the technology and market trends from period to period. 

### 4.2. Layer Mapping

#### 4.2.1. Classifying the Data into Layers Based on BERT

In the second module, the abstracts are separately extracted from the tech-related and market-related data of TS_1_, TS_2_, and TS_3_ for text analysis. After that, we pre-processed the data extracted from TS_1_, TS_2_, and TS_3_ by dividing the 11,671 abstracts into 85,656 individual sentences with Python’s NLTK package. We then conducted a BERT model to classify technical and market data. Even if BERT is beneficial for classifying technical and market data, how to build a training set cannot do without the help of experts. Three experts engaged in introducing hyperuricemia drugs for more than ten years were invited to construct the training set. With the help of the domain experts, we reviewed the data sentence-by-sentence and extracted 30 percent sentience randomly from the entire data set as the training set. The extracted training set is then manually labeled and classified into technical-related, market-related, and irrelevant data, denoted by C_1_ and C_2_, and C_0_, respectively.

The BERT model is pre-trained, and each subset (TS_1_, TS_2_, TS_3_) was divided into three categories. This study calls for technical and market-related data in developing a data-driven technology roadmap. The irrelevant data in C_0_ is useless to our research [15]. Thus, after classification, we only kept 54,026 tech-related sentences in C_1_ and 31,630 market-related sentences in C_2_. Finally, the whole dataset is divided into six subsets and classified into different layers of data-driven TRM. C_1_ in TS_1_ means tech-related data in 2010–2013, C_1_ in TS_2_ means tech-related data in 2014–2018, C_1_ in TS_3_ means tech-related data in 2019–2021, C_2_ in TS_1_ means market-related data in 2010–2013, C_2_ in TS_2_ means market-related data in 2014–2018, and C_2_ in TS_3_ means market-related data in 2019–2021.

#### 4.2.2. Semantic Analysis for the Technology Layer and Market Layer Based on SAO

After classifying the data into different layers of data-driven TRM, we chose SAO to extract technical and market semantic structures, reflecting the contexts. The SAO structures are retrieved from each sentence employing word annotation. The word annotation is performed using the Stanford parser, employed by many investigators. In reliance on the Stanford parser, 54,026 technology SAO structures and 31,630 market SAO structures were extracted in the timeframe. The original SAOs are heavily redundant. To analyze them efficiently, we filtered, cleaned, and combined the SAOs. We then deleted 1022 duplicates of technology SAO structures and 1681 market structures. A total of 41,826 unique technology SAO structures and 10,621 unique market SAO structures remained. We are removing the 35,483 invalid technology SAO records and 9545 market SAO records with no subjects, actions, or objects. There were 6343 technology structures and 1076 market structures left, as shown in Table 2 and Table 3.

### 4.3. Contents Mapping

#### 4.3.1. Pre-Processing the SAO Components

In the third module, we extract technology and market SAO components topics in the timeframe. Thus, we divided the 6343 technology SAOs and 1076 market SAOs into 19 subsets, as shown in Table 4. For example, T-S-TS_1_ meant the technology S components related data in 2010–2013, and M-S-TS_1_ represents the market S components associated data in 2010–2013. To remove the data noise, we pre-processed the 18 sub-datasets group by group, such as word token, stemming, lemmatization, and excluding stop words. In addition to the 972 basic stop-words, we designated 2188 domain-specific stop-words and excluded them from analysis, such as ‘uric acid’, ‘hyperuricemia’, ‘treatment’, etc.

#### 4.3.2. Identify Topics of SAO Components for Technology and Market Layers Based on BTM

We conducted a BTM-based topic model to identify meaningful core and potential technology and market topics automatically. To determine the optimal number of topics, we evaluate the value of coherence while varying the number of topics. The maximization of the coherence value defines the number of optimal topics. We then calculated the coherence value subset by subset, as shown in Table 5, Figure 5 and Figure 6.

As a result, 9, 1, 10, 10, 2, 10, 9, 1 and 9 topics were derived for sub-set T-S-TS_1_, T-A-TS_1_, T-O-TS_1_, T-S-TS_2_, T-A-TS_2_, T-O-TS_2_, T-S-TS3, T-A-TS3, and T-O-TS3. 9, 1, 7, 9, 1, 2, 10, 1, and 2 topics were derived for sub-set M-S-TS1, M-A-TS1, M-O-TS1, M-S-TS2, M-A-TS2, M-O-TS2, M-S-TS3, M-A-TS3, and M-O-TS3, respectively. The critical and potential topics of SAO components for technology and market layers are shown in Appendix A. Take Appendix A as an example. The topics identified can be defined as O components for the market layer in the time-based framework. There are topics in stage TS_1_ such as key amino acid mutations (M-O-TS_1_-T_1_), reduced side effects of small molecule drugs (M-O-TS_1_-T_2_), Chinese medicine treatment (M-O-TS_1_-T_3_), the need to develop drugs with low nephrotoxicity (M-O-TS_1_-T_4_), the need to develop immunogenic drugs with low (M-O-TS_1_-T_5_), drug market risk (M-O-TS_1_-T_6_), improve patients’ quality of life (M-O-TS_1—_T_7_). There are small molecule drug products such as Zurampic and Duzallo (M-O-TS_2_-T_1_); enterprises need to find key amino acid sites (M-O-TS_2_-T_2_) topics in stage TS_2_. There are reduced drug side effects (M-O-TS_3_-T_1_) and market demand for drugs with low immunogenicity (M-O-TS_3_-T_2_) topics in stage TS_3_. Identifying SAO component topics at the technical and market layers helps each investigator understand the complex topics in his field and throughout hyperuricemia drug development.

### 4.4. Opportunity Finding

#### 4.4.1. Identify Potential Connections Based on Link Prediction

The last section forecasts possible technical opportunities for hyperuricemia. We constructed the entire network based on link prediction. To build the whole network, we selected the data set pre-processed in Section 4.3.1 to train the model in technology and market layers based on link prediction. The entire technology network consists of 7693 nodes and 17,190 edges. The whole market network consists of 1868 nodes and 2907 edges. 

#### 4.4.2. Integrating TRM and Analyzing Opportunities

We select some nodes and edges in the technology and market sub-networks for visualization. Only the nodes and edges that are potentially connected are selected for visualization from the 61 technology topics and 42 market topics extracted in Section 4.3.2.

Next, in the timeframe, we arrange the visualizations in technology and market layers of data-driven TRM from sub-layer S to sub-layer O, as shown in Figure 3 and Figure 7. Finally, we divided the communities. The technical layer was divided into six communities represented by C_1_, C_2_, C_3_, C_4_, C_5_, and C_6_. The marketing layer was divided into five communities represented by C_7_, C_8_, C_9_, C_10_, and C_11_.

Figure 3 and Figure 7 integrate semantic analysis, topic modeling, and link prediction results and show the final technology roadmap. We analyzed the 11 communities in the technology and market layers. Several technological opportunities can be identified. The technical opportunities in the C_1_, C_4_, C_7,_ and C_10_ communities are mostly related to chemical-based drugs such as small molecule drugs. The technological opportunities in the C_2_, C_3_, C_6_, C_8_, and C_11_ communities are primarily associated with biological-based medications such as protein drugs.

Take protein drugs related communities as an example. The technology opportunities for hyperuricemia drugs are protein drugs related to the C_2_, C_3_, C_6_, C_8_, and C_11_ communities. From 2010 to 2013, the opportunities focus on how to extend the half-life of protein drugs in patients and how to develop new protein drugs. Protein drug research is mainly based on recombinant Aspergillus flavus uricase and PEG-modified urate oxidase derived from pig baboons (T-S-TS_1_-T_5_ in C_2_, M-S-TS_1_-T_7_ in C_8_). All uric acid oxidases on the market are foreign proteins (M-O-TS_1_-T_4_ in C_11_, M-O-TS_1_-T_5_ in C_11_). It is easy to produce antibodies after long-term administration of foreign protein. The immunogenicity of uric acid oxidase limits its use, and there are many adverse reactions after entering the market (T-S-TS_1_-T_1_ in C_2_). From 2014 to 2018, it was discovered that active urate oxidase drugs with relatively low immunogenicity could be obtained by “resurrecting” human urate oxidase (M-O-TS_2_-T_2_ in C_11_). And it ultimately achieve the purpose of treating hyperuricemia and gout (T-S-TS_2_-T_1_, T-S-TS_2_-T_2_, T-S-TS_2_-T_4_, and T-S-TS_2_-T_5_ in C_2_, T-A-TS_2_-T_1_ in C_4_). From 2019 to 2021, while trying to restore human urate oxidase activity, researchers continued to study how to reduce the half-life of existing protein drugs. We can derive urate oxidase from Aspergillus flavus (T-S-TS_3_-T_1_ in C_2_, T-S-TS_3_-T_4_, and T-S-TS_3_-T_5_ in C_2_). How to eliminate or reduce the immunogenicity of existing urate oxidase and obtain active and non-immunogenic human urate oxidase drugs will be the future direction of protein drug development. At these three stages, developers’ attention to restoring human uric acid oxidase activity is greater than the market demand. In the future, a comparison of DNA sequences of uric acid oxidase from different sources could be considered to discover the critical amino acid sites that affect the enzyme activity (T-S-TS_2_-T_1_ in C_2_). The essential amino acid sites were then mutated (M-O-TS_2_-T_2_ in C_2_). After each completed mutation, human uric acid oxidase (T-A-TS_2_-T_2_ in C_4_) was induced, and the activity of uric acid oxidase (T-S-TS_3_-T_1_ in C_2_) was assayed after affinity purification. We can resurrect the human uric acid oxidase pseudogene through the above pathway. We can also overcome the disadvantage that existing oxidase drugs are immunogenic and produce antibodies when used for a long time. We can obtain human uric acid oxidase with high drug activity but low immunogenicity to improve the treatment of hyperuricemia and gout.

The technology opportunities for hyperuricemia drugs are small molecule drugs related to the C_1_, C_4_, C_7_, and C_10_ communities. Similarly, how to reduce the side effects of small molecule drugs will be the future direction of small molecule drug R&D, such as lowering hepatotoxicity and nephrotoxicity(M-O-TS_1_-T_2_). At these three stages, developers’ attention to reducing the side effects of small molecule drugs is lower than the market demand. Reducing the side effects of small molecule drugs is critical in the future. We can try to discover new structures from Chinese medicine or in combination with Chinese medicine, such as combining heat-clearing, dampness-relieving herbs with small molecule drugs (M-S-TS_3_-T_10_ in C_7_, M-S-TS_3_-T_10_ in C_3_). In addition, it would be good to try to change diet patterns combined with small molecule drug therapy. We can reduce the intake of high purine foods and prevent a eutrophication diet (T-S-TS_2_-T_6_ in C_1_, T-O-TS_3_-T_2_ in C_5_, M-S-TS_2_-T_9_ in C_7_, M-S-TS_3_-T_7_ in C_8_).

## 5. Discussion 

This paper presents a systematic approach to developing a data-driven TRM to identify potential technology opportunities for hyperuricemia drugs. Despite the contributions of previous research, we can extend the existing data-driven TRM from three aspects. These are identifying potential opportunities with data-driven TRM, the process, and the data source of data-driven TRM.

Compared with current trends, we chose link prediction from the perspective of technology forecasting to identify potential technology opportunities for opportunity finding of data-driven TRM. SAO considers existing technology connections, while technology opportunities are often hidden in potential relationships. SAO needs to be combined with link prediction to predict technology opportunities better. Therefore, we identify possible links between unconnected nodes based on link prediction. Based on the results of the link predictions, we should focus on resurrecting the pseudogene of human uric acid oxidase and reducing the toxicity of small molecule drugs in the future.

From the perspective of the data process, compared with keyword-based analysis, we chose SAO to extract critical technology information for layer mapping of data-driven TRM reflecting the contexts. The SAO structure was widely used to analyze documents such as patents, online review demand extraction, and paper. Although SAO is commonly used in TOA, the SAO structures must be refined for efferent analysis. Therefore, this article identifies topics of SAO components for different layers of data-driven TRM based on BTM. It can reduce the redundancy of SAO effectively. Besides that, it can also help to extract which areas the technical solution focuses on, how the solution works, and which parts of the solution are the targets of SAOs. Based on the potential opportunities identified by the link prediction, the realization path of the opportunity is inferred by the SAO structure. For example, it is critical to resurrect the pseudogene of human uric acid oxidase. We could consider comparing DNA sequences of different sources of uric acid oxidase to discover the critical amino acid site that affects enzyme activity (T-S-TS_2_-T_1_ in C_2_). We would then mutate the essential amino acid site (M-O-TS_2_-T_2_ in C_2_) and induce the expression of uric acid oxidase (T-A-TS_2_-T_2_ in C_4_) affinity purification to analyze the enzyme activity (T-S-TS_3_-T_1_ in C_2_).

From the perspective of data source selection, compared with selecting patents as technical data and commercial reports as market data, we choose patent, literature, and commercial reports as technical and market data of data-driven TRM. We distinguish technology and market data from the vast amount of literature, patent, and commercial data automatically based on BERT and combined with the small domain training set to train BERT models to classify hyperuricemia drugs with high accuracy. To identify potential technology opportunities in concert with market demand, it is necessary to analyze technology opportunities from both market and technology perspectives.

## 6. Conclusions

It is essential to assist hyperuricemia developers via a data-driven TRM for TOA. However, less attention has been paid to integrating multiple analytical tools within a data-driven TRM to identify potential technology opportunities automatically, such as SAO, BTM, and link prediction. This study extends the existing data-driven TRM from several aspects to fill this gap. There are data process, technology forecasting, and data source selection. And we try to respond to the following questions in this study. First, we illustrate how to build a semantic-based data-driven TRM and identify topics of SAO components based on BTM. Second, we point out how to identify potential technology opportunity points through link prediction. Last, we demonstrate how to extract technical and market information automatically based on text classification tools from various patents, literature, and business data. Therefore, this research provides an attractive option for hyperuricemia drugs TOA. It is critical to narrow down the technology research topics, reducing R&D risks, and supporting the decision-making of hyperuricemia drug research.

Despite the promise of data-driven TRM in hyperuricemia drugs’ TOA, challenges have remained in one aspect that can be addressed by future research. First, the SAO structure is complex. It is challenging for machine learning models to identify semantic relations from complex sentences. In future research, we will focus on exploring how to extract SAO structures from complex sentences in the future. Second, patent, literature, and commercial data are not real-time data because of the time lag. It is difficult to obtain the latest technology and market information for TOA. In the future, we will consider developing a dynamic data-driven TRM, such as by adopting dynamic topic models for the topic analysis of the SAO structure. Finally, this paper selects patent, technical and commercial data for data source selection. In the future, more diversified data sources, such as online reviews, drug instructions, etc., will be selected to complement the existing analysis effectively.

## Figures and Tables

**Figure 1 pharmaceuticals-15-01357-f001:**
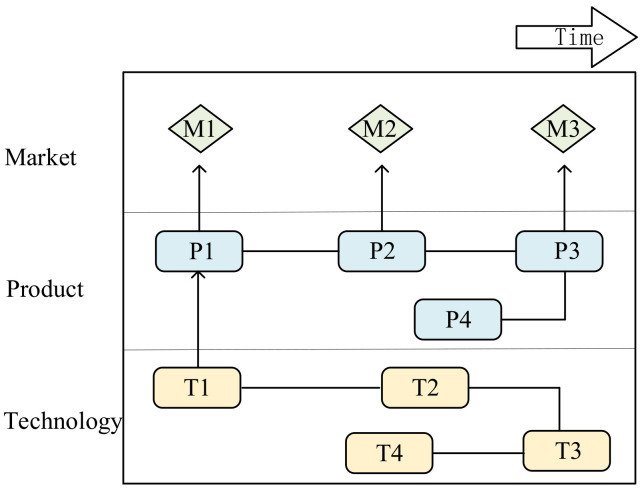
Structure of technology roadmap.

**Figure 2 pharmaceuticals-15-01357-f002:**
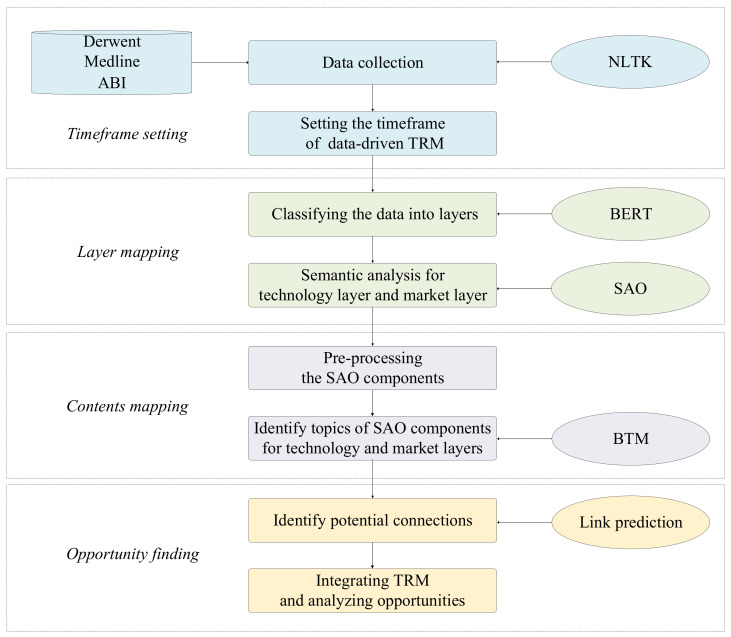
Research framework.

**Figure 3 pharmaceuticals-15-01357-f003:**
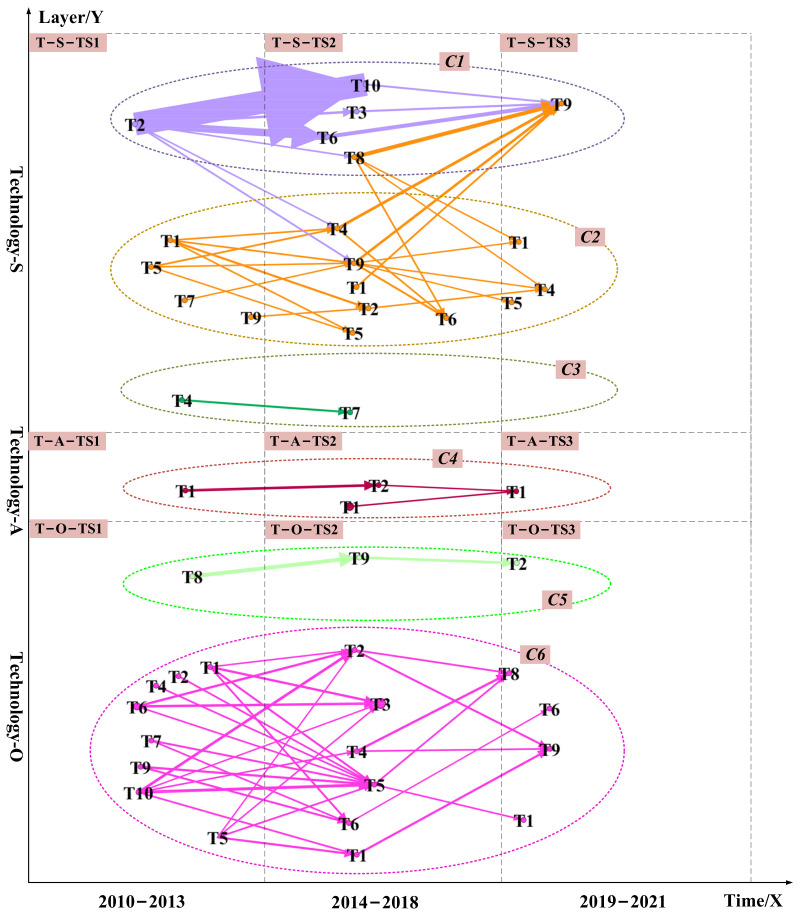
The data-driven TRM for the technology layer. The technical layer was divided into six communities represented by C_1_, C_2_, C_3_, C_4_, C_5_, and C_6_. Dashed cycles with different colors highlight diverse communities. The edges’ width and arrows’ width represent the probability of a potential link between unconnected nodes. The wider the edges and arrows, the higher the likelihood of a potential link. Arrows and edges with different colors represent different community themes.

**Figure 4 pharmaceuticals-15-01357-f004:**
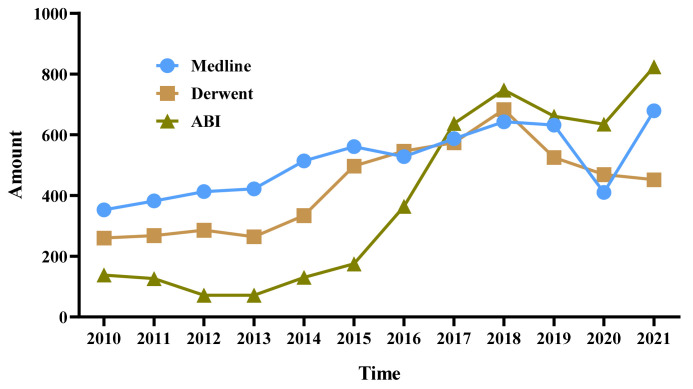
Statistical results of papers, patents, and commercial data related to hyperuricemia drugs.

**Figure 5 pharmaceuticals-15-01357-f005:**
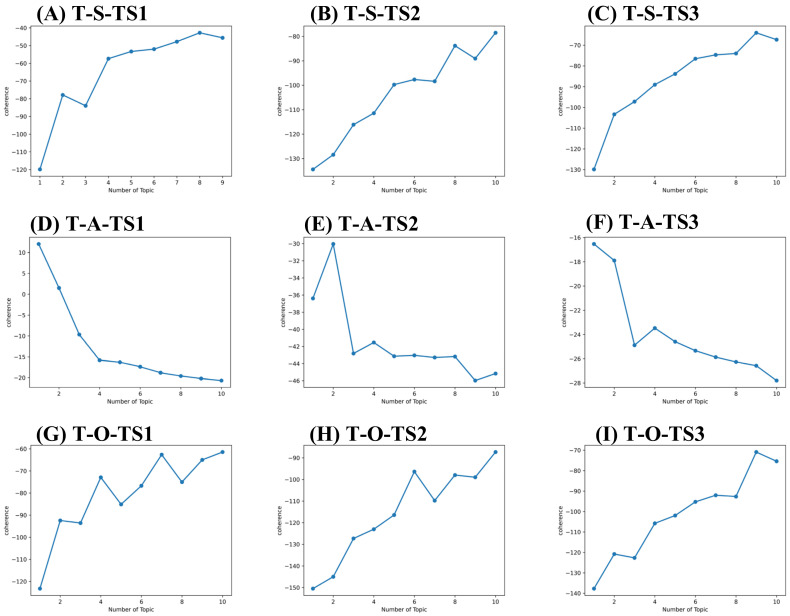
Topic coherence curve for technology layer. (**A**) Topic coherence value of S components for technology layer in 2010–2013 (T-S-TS1). (**B**) Topic coherence value of S components for technology layer in 2014–2018 (T-S-TS2). (**C**) Topic coherence value of S components for technology layer in 2019–2021 (T-S-TS3). (**D**) Topic coherence value of A components for technology layer in 2010–2013 (T-A-TS1). (**E**) Topic coherence value of A components for technology layer in 2014–2018 (T-A-TS2). (**F**) Topic coherence value of A components for technology layer in 2019–2021 (T-A-TS3). (**G**) Topic coherence value of O components for technology layer in 2010–2013 (T-O-TS1). (**H**) Topic coherence value of O components for technology layer in 2014–2018 (T-O-TS2). (**I**) Topic coherence value of O components for technology layer in 2019–2021 (T-O-TS3).

**Figure 6 pharmaceuticals-15-01357-f006:**
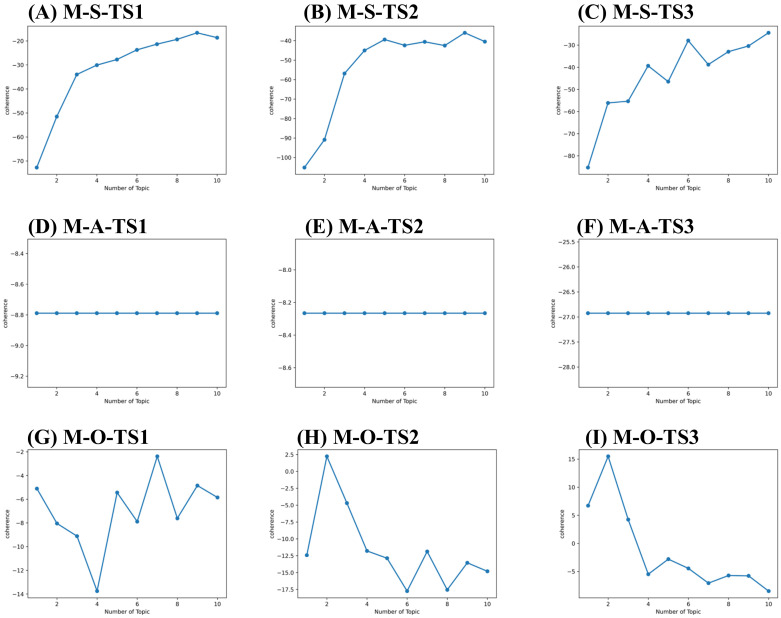
Topic coherence curve for market layer. (**A**) Topic coherence value of S components for market layer in 2010–2013 (M-S-TS1). (**B**) Topic coherence value of S components for market layer in 2014–2018 (M-S-TS2). (**C**) Topic coherence value of S components for market layer in 2019–2021 (M-S-TS3). (**D**) Topic coherence value of A components for market layer in 2010–2013 (M-A-TS1). (**E**) Topic coherence value of A components for market layer in 2014–2018 (M-A-TS2). (**F**) Topic coherence value of A components for market layer in 2019–2021 (M-A-TS3). (**G**) Topic coherence value of O components for market layer in 2010–2013 (M-O-TS1). (**H**) Topic coherence value of O components for market layer in 2014–2018 (M-O-TS2). (**I**) Topic coherence value of O components for market layer in 2019–2021 (M-O-TS3).

**Figure 7 pharmaceuticals-15-01357-f007:**
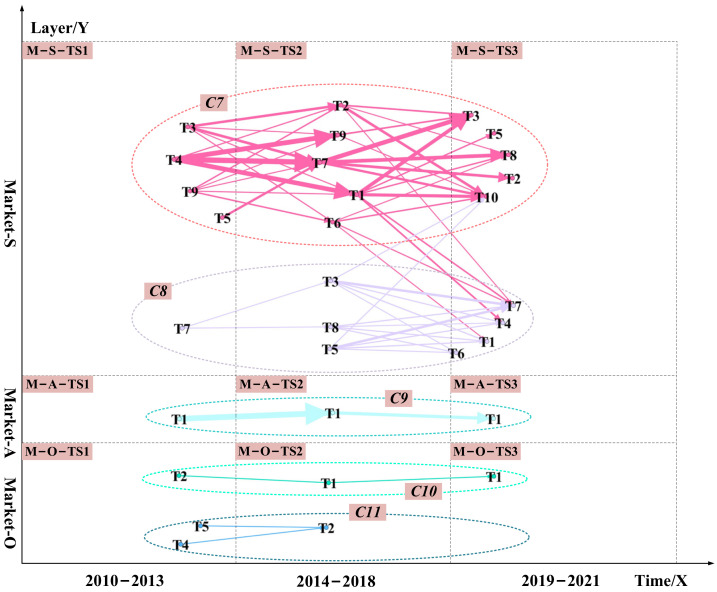
The data-driven TRM for the market layer. The marketing layer was divided into five communities represented by C_7_, C_8_, C_9_, C_10_, and C_11_. Dashed cycles with different colors highlight diverse communities. The edges’ width and arrows’ width represent the probability of a potential link between unconnected nodes. The wider the edges and arrows, the higher the likelihood of a potential link. Arrows and edges with different colors represent different community themes.

**Table 1 pharmaceuticals-15-01357-t001:** Database for data-driven TRM.

Type	Data Source	Retrieval Strategy	Count
Paper	Medline	MH = (gout OR hyperuricemia)	6124
Patent	Derwent	IP = (A61P-019/06)	5158
Market	ABI	hyperuricemia	4582

**Table 2 pharmaceuticals-15-01357-t002:** Number of SAOs for the technology layer.

Time Series	Sentences	SAO Components			
		Original	Duplicate	Incomplete	Reserved
2010–2013	12,161	9912	202	8263	1447
2014–2018	24,441	19,902	596	16,408	2898
2019–2021	17,424	13,034	224	10,812	1998
In total	54,026	42,848	1022	35,483	6343

**Table 3 pharmaceuticals-15-01357-t003:** Number of SAOs for market layer.

Time Series	Sentences	SAO Components			
		Original	Duplicate	Incomplete	Reserved
2010–2013	4371	2439	219	1960	260
2014–2018	17,799	6315	1035	4767	513
2019–2021	9460	3548	427	2818	303
In total	31,630	12,302	1681	9545	1076

**Table 4 pharmaceuticals-15-01357-t004:** The group result of SAO components for the technology and market layer.

Layer	SAO Components	Time Series	Group
Technology	S	2010–2013	T-S-TS_1_
		2014–2018	T-S-TS_2_
		2019–2021	T-S-TS_3_
	A	2010–2013	T-A-TS_1_
		2014–2018	T-A-TS_2_
		2019–2021	T-A-TS_3_
	O	2010–2013	T-O-TS_1_
		2014–2018	T-O-TS_2_
		2019–2021	T-O-TS_3_
Market	S	2010–2013	M-S-TS_1_
		2014–2018	M-S-TS_2_
		2019–2021	M-S-TS_3_
	A	2010–2013	M-A-TS_1_
		2014–2018	M-A-TS_2_
		2019–2021	M-A-TS_3_
	O	2010–2013	M-O-TS_1_
		2014–2018	M-O-TS_2_
		2019–2021	M-O-TS_3_

**Table 5 pharmaceuticals-15-01357-t005:** Topic coherence value for each sub-set.

Technology			Market		
Group	Number of Topics	Coherence	Group	Number of Topics	Coherence
T-S-TS_1_	9	−45.59	M-S-TS_1_	9	−16.61
T-A-TS_1_	1	12.03	M-A-TS_1_	1	−8.79
T-O-TS_1_	10	−61.45	M-O-TS_1_	7	−2.38
T-S-TS_2_	10	−78.55	M-S-TS_2_	9	−35.95
T-A-TS_2_	2	−30.04	M-A-TS_2_	1	−8.27
T-O-TS_2_	10	−87.32	M-O-TS_2_	2	2.24
T-S-TS_3_	9	−64.03	M-S-TS_3_	10	−24.46
T-A-TS_3_	1	−16.53	M-A-TS_3_	1	−26.92
T-O-TS_3_	9	−70.89	M-O-TS_3_	2	15.48

## Data Availability

Data is contained within the article and Appendix A.

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
