# Peer review of "Data-Driven Technology Roadmaps to Identify Potential Technology Opportunities for Hyperuricemia Drugs"

_pharmaceuticals, 2022, doi:10.3390/ph15111357_

Round 1

Reviewer 1 Report

The authors explored technology roadmap (TRM) approach to identify potential technology opportunities. The state of the art was nicely described to build up the hypothesis and objective is the study. The introduction and methodology are well written. I have the following concerns, especially in the results sections.

Line 185-186: Briefly expand the outcomes of these studies to understand previous developments in the domain.

Line 268: In the paragraph, it would be interesting to see the limitations of BTM in the current context.

Line 469: What is the logic here behind dividing data into six subsets?

Figure 4: x and y scales are not the same for each plot. I guess the comparison will be more realistic while the scale will be the same. At least the scale should be the same in each horizontal panel.

Table 6-11: I think Table 6 to 11 represents the topic results of various components for the market layer. I think this information should be in the supplementary material.

Figure 5 and 6: The figures are not self-explanatory. The color codes and thickness of the arrows should be defined in the legend. Authors may also include brief information about clusters.  

Author Response

Dear Editor and reviewers,

We appreciate your favorite consideration and the three reviewers' insightful comments concerning our manuscript entitled "Identify potential technology opportunities for Hyperuricemia drugs based on data-driven technology roadmaps" (Manuscript ID:pharmaceuticals-1929159). Those comments are precious and helpful for improving the quality and readability of our paper, as well as the crucial guiding significance to our future research. We have carefully studied the comments and revised the paper according to the reviewers' comments. We hope this revision can meet with approval. The major modifications and additional analysis undertaken are as follows:

  • Checked that all references were relevant to the manuscript's contents.
  • Rewrite the abstract.
  • Rewrite the first paragraph of the introduction.
  • Briefly expanded on the previous developments in the domain.
  • Added the limitations of BTM in the current context.
  • Changed in the result and Figured as follows:
  • Explain the logic behind dividing data into six subsets.
  • Moved Table 6 to 11 in the original manuscript to the supplementary material.
  • Modified Figure 4 in the original text.
  • Introduced the color codes and the thickness of the arrows and gave brief information about clusters in the legend in Figure 5 and 6.
  • Improved the writing style and research presentation as follows:
  • Combined the short sentences (4-5 words) using appropriate connecting words.
  • Used the different connecting words in subsequent sentences.
  • Modified improper sentences.

Detailed responses to the reviewers' comments are provided in a separate document titled "Detailed response to reviewers' comments." The critical revised material in the revised version of the manuscript are marked in blue and our responses are marked in red.

We extend our appreciation to the Editor and the reviewers.

Sincerely,

Jinfeng Wang

Reviewer 2 Report

The writing style and research presentation should be improved significantly.

Some examples on the writing style that must be improved:

First paragraph of introduction needs to be rewritten.

Don't use the same connecting word in subsequent sentences -- "when" on L38 and on L40

L108 better to say this article is proposing

L108 why data is capitalized

Many sentences all over the manuscript are very short (4-5 words) -- some should be combined together using suitable connecting words to make more informative sentences. 

Author Response

Dear Editor and reviewers,

We appreciate your favorite consideration and the three reviewers' insightful comments concerning our manuscript entitled "Identify potential technology opportunities for Hyperuricemia drugs based on data-driven technology roadmaps" (Manuscript ID:pharmaceuticals-1929159). Those comments are precious and helpful for improving the quality and readability of our paper, as well as the crucial guiding significance to our future research. We have carefully studied the comments and revised the paper according to the reviewers' comments. We hope this revision can meet with approval. The major modifications and additional analysis undertaken are as follows:

Checked that all references were relevant to the manuscript's contents. 
Rewrite the abstract. 
Rewrite the first paragraph of the introduction. 
Briefly expanded on the previous developments in the domain. 
Added the limitations of BTM in the current context. 
Changed in the result and Figured as follows: 
Explain the logic behind dividing data into six subsets. 
Moved Table 6 to 11 in the original manuscript to the supplementary material. 
Modified Figure 4 in the original text. 
Introduced the color codes and the thickness of the arrows and gave brief information about clusters in the legend in Figure 5 and 6. 
Improved the writing style and research presentation as follows: 
Combined the short sentences (4-5 words) using appropriate connecting words. 
Used the different connecting words in subsequent sentences. 
Modified improper sentences. 

Detailed responses to the reviewers' comments are provided in a separate document titled "Detailed response to reviewers' comments." The critical revised material in the revised version of the manuscript are marked in blue and our responses are marked in red.
We extend our appreciation to the Editor and the reviewers.

Sincerely,
Jinfeng Wang

Reviewer 3 Report

Manuscript title:  Data-Driven Technology Roadmaps to Identify Potential Tech-2 nology Opportunities for Hyperuricemia Drugs

 Journal:     Pharmaceuticals (ISSN 1424-8247)

Manuscript ID :  pharmaceuticals-1929159

Dear Editor: 

Thank you to send me this manuscript. I wrote my evlution below:

(1) Present original findings, conclusions or analysis that has not been published previously by the authors or others : Yes

  (2) Written clearly: No

 (3) Have a high impact in its subfield: Yes

This manuscripte (Data-Driven Technology Roadmaps to Identify Potential Tech-2 nology Opportunities for Hyperuricemia Drugs). The work presented here is interesting and scalable. It is a very relevant subject of study. I would recommend publication of this manuscript in this journal after the authors rigorously addressed some comments listed below.

1)      The abstract section does not have all important information. I prefer to rewrite this section by fiiling the spesefice and importante informations. So, it has more confuse for me.

2)      The introduction section has a very good information, the authors do the best for this section.

3)        There are several passages of the manuscript which are incorrect.

Finally, if the authors do all these comments, I recommend to publish this manuscript in this journal.

        Best Regards

Assist. Pro. Dr.Mohammed H. Mohammed

Author Response

(The authors gave the same response as above.)

Round 2

Reviewer 1 Report

Thanks for considering the suggestions and improving the text. I agree with the revisions made by authors.